# Electrical manipulation of skyrmions in a chiral magnet

Weiwei Wang [1,2], Dongsheng Song[1 ✉], Wensen Wei[2], Pengfei Nan[1], Shilei Zhang[3], Binghui Ge[1], Mingliang Tian [2,4,5], Jiadong Zang [6,7,8 ✉] & Haifeng Du[1,2,4 ✉]

Writing, erasing and computing are three fundamental operations required by any working electronic device. Magnetic skyrmions could be essential bits in promising in emerging topological spintronic devices. In particular, skyrmions in chiral magnets have outstanding properties like compact texture, uniform size, and high mobility. However, creating, deleting, and driving isolated skyrmions, as prototypes of aforementioned basic operations, have been a grand challenge in chiral magnets ever since the discovery of skyrmions, and achieving all these three operations in a single device is even more challenging. Here, by engineering chiral magnet $Co_8Zn_{10}Mn_2$ into the customized micro-devices for in-situ Lorentz transmission electron microscopy observations, we implement these three operations of skyrmions using nanosecond current pulses with a low current density of about $10^{10}$ A·m$^{-2}$ at room temperature. A notched structure can create or delete magnetic skyrmions depending on the direction and magnitude of current pulses. We further show that the magnetic skyrmions can be deterministically shifted step-by-step by current pulses, allowing the establishment of the universal current-velocity relationship. These experimental results have immediate significance towards the skyrmion-based memory or logic devices.

[1] Institutes of Physical Science and Information Technology, Anhui University, Hefei 230601, China. [2] Anhui Province Key Laboratory of Condensed Matter Physics at Extreme Conditions, High Magnetic Field Laboratory, HFIPS, Anhui, Chinese Academy of Sciences, Hefei 230031, China. [3] School of Physical Science and Technology, ShanghaiTech University, Shanghai 201210, China. [4] Science Island Branch of Graduate School, University of Science and Technology of China, Hefei, Anhui 230026, China. [5] School of Physics and Materials Science, Anhui University, Hefei 230601, China. [6] Department of Physics and Astronomy, University of New Hampshire, Durham, NH 03824, USA. [7] Materials Science Program, University of New Hampshire, Durham, NH 03824, USA. [8] Institute for Theoretical Physics, University of Cologne, 50937 Cologne, Germany. ✉email: dsong@ahu.edu.cn; Jiadong.Zang@unh.edu; duhf@hmfl.ac.cn

The skyrmion lattice, a new magneto-crystalline order composed of topologically stable nanometer-sized magnetic whirls, was discovered in a chiral magnet MnSi a decade ago[1,2]. One year later, subsequent experiments showed that the skyrmion lattice could be driven to motion using direct currents (DC) at an ultra-low current density[3,4]. This prominent feature, together with its topological stability and small size, opens the door to skyrmion-based spintronic devices[5], such as racetrack memory[6], logic devices[7], and neuromorphic computation[8,9]. However, from the technological point of view, the controllable creation, deletion, and motion of isolated skyrmions[10–12], rather than skyrmion lattices, using nanosecond current pulses are of critical significance in practical applications[11,13,14]. These operations of the Néel skyrmions induced by the interfacial Dzyaloshinskii-Moriya interaction[15] (DMI) have been demonstrated in heterostructures using the spin-orbit torque (SOT)[13,14,16–20]. However, the same operations of skyrmions using spin-transfer-torque (STT) in bulk chiral magnets have progressed very slowly in the past decade. In contrast to the magnetic thin films, the main challenge is fabricating high-quality microdevices from bulk samples for in-situ magnetic imaging. Very few experiments of electrical manipulation of isolated skyrmions[12,21] have been reported, but long and wide current pulses (milliseconds) had to be used.

Here, we present systematic experiments on the STT-induced creation, deletion, and motion of magnetic skyrmions in chiral magnet $Co_8Zn_{10}Mn_2$[22] at room temperature. Under a relatively low current density, the creation and deletion of isolated skyrmion have been realized via a geometrical notch at the device edge. Moreover, the universal current-velocity relation of skyrmions motions is established, and a combined operation of creation, motion, and deletion is implemented.

## Results

### Skyrmions creation and deletion with a Notch.
We first demonstrate the current induced skyrmion creation using a notch at the sample boundary[23]. The fabricated micro-device[24] comprises two Pt electrodes and a thin lamella with a thickness of ~150 nm (see Methods, Supplementary Fig. S1 and Movie 1). A $190 \times 280$ nm$^2$ notch is specifically designed to serve as a nucleation seed for creating skyrmions using current. The notch width of ~190 nm is comparable to the period of the spin helix ($L \sim 114$ nm)[25], much smaller than that reported in a recent FeGe-based device[12], making the creation of a single skyrmion possible.

Figure 1a shows the snapshots of the skyrmion creation process after applying a sequence of current pulses with the width of 20 ns and current density of $-4.26 \times 10^{10}$ A $\cdot$ m$^{-2}$ in the $x$-direction (see the details in Supplementary Movie 2). Initially, the sample is in the conical state under the external field $B$ of 70 mT. After applying two pulses, one skyrmion with a topological charge of $Q = -1$ is created. The created skyrmion is attached to the notch, indicating the attractive interaction between the skyrmion and the notch. Further application of current pulses continuously creates skyrmions one-by-one till the end of the 12th pulse (Fig. 1b). After that, it occasionally happens that no skyrmion is created under a few applications of current pulses. Nevertheless, the linear relationship between the number of created skyrmion and applied pulses is well identified (Fig. 1b and see also Supplementary Fig. S2 for other datasets). At last, a skyrmion cluster composed of 19 skyrmions is created after the 21st pulse. It stretches into a ribbon-like shape due to the skyrmion Hall effect[16,19,26].

The spin textures of the notch play an essential role in creating the skyrmions. The notch has a sizeable in-plane magnetization component that is perpendicular to the direction of electrical current, while it is parallel at the normal edges. Under the electrical current, the spin textures could swell out and then nucleate a skyrmion due to the STT, but absent at the regular edges (Supplementary Fig. S3). Therefore, the skyrmion nucleation energy is lower at the notch than the "normal" sample edges.

The effect of the current pulse magnitude and direction on the skyrmion creation is summarized in Fig. 1c, where the parameter $<N_s>$ represents the average number of created skyrmions per current pulse. Under a negative current pulse, the threshold current density for the skyrmion creation is approximate $-3.4 \times 10^{10}$ A $\cdot$ m$^{-2}$, which is one order of magnitude smaller than the theoretical estimation[23]. On the contrary, a positive current with the same current density failed to create skyrmions (Fig. 1d), indicating the asymmetric STT effects with regard to the current direction[23]. This asymmetry of STT-induced skyrmion creation originates from the breakdown of reflection symmetry due to the unique direction of the spin precession[23]. Consequently, skyrmions with $Q = +1$ could only be created with a positive current and a negative field (Fig. 1d). The thermal effect starts to dominate the creation process when the current density exceeds $5 \times 10^{10}$ A $\cdot$ m$^{-2}$. After that, the Joule heating becomes increasingly prominent. The ultrafast field-warming beyond the $T_c$ and then field-cooling process results in the creation of skyrmions[27,28]. As a result, the unidirectionality is weakened (Supplementary Movie 3) and $<N_s>$ rapidly increases (Fig. 1c and Supplementary Fig. S4).

The unidirectionality of skyrmion creation with the notch allows us to delete the skyrmion by using its inverse process. Figure 2a shows the representative Lorentz-TEM images of an $N_s = 15$ skyrmion cluster after successive applications of negative current pulses under $j \sim 4.06 \times 10^{10}$ A $\cdot$ m$^{-2}$ and $B \sim 70$ mT (see the details in Supplementary Movie 4). Once the positive current pulses are applied, the number of skyrmions $N_s$ decrease quickly with the increase of pulse number. Finally, only one skyrmion is left. It is attached to the edge owing to the attractive interaction between skyrmion and the edge[29], which is different from the two-dimensional system where the boundary twist owing to DMI induces a repulsive potential to skyrmions.

The number of remaining skyrmions as a function of current pulses is shown in Fig. 2b. The average number of deleted skyrmions per pulse depends on the strength of current density, as shown in Fig. 2c. The deletion rate increases with the current density (Supplementary Fig. S5) and reaches its maximum at $j \sim 3.65 \times 10^{10}$ A $\cdot$ m$^{-2}$. Note that the threshold current density required to delete skyrmions is much smaller than that to create skyrmions, although it appears to be the inverse process of the latter. It can be attributed to the asymmetric energy landscape between the conical state and the skyrmion state. The energy barrier of skyrmion deletion is smaller than that of skyrmion creation[30]. In principle, a flat edge should also absorb skyrmions due to the inevitable skyrmion Hall effect under large current density[16]. However, it did not occur even at $j \sim 4.06 \times 10^{10}$ A $\cdot$ m$^{-2}$ (Supplementary Fig. S6), indicating the crucial role of the rectangular notch.

### Skyrmions motion by current.
We now turn to the motion of skyrmions driven by STT. The universal current-velocity relation of skyrmion dynamics under STT has been theoretically addressed[31]. The longitudinal velocity of a skyrmion is derived as $v_x \approx -bj$ under the electrical current, where $b$ is a constant and $j$ is the current density (see Supplementary Note I). The experimental results of the nanosecond-pulse-driven skyrmions motion are summarized in Figs. 3–5 with varied skyrmion numbers. For $Q = -1$, the skyrmion moves along the $+x$

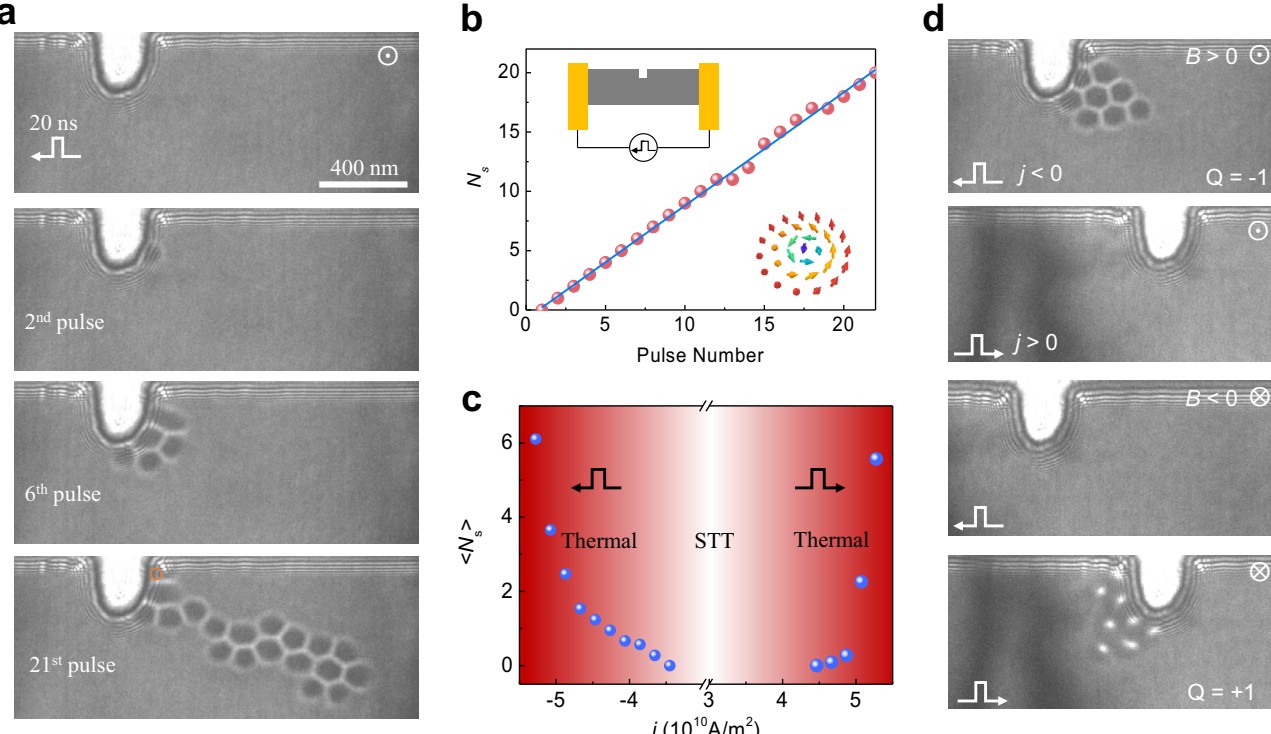

**Fig. 1 Skyrmion creation with a notch. a** A sequence of Lorentz TEM images of the skyrmion creation process after applying designated numbers of current pulses. After the 2nd, 6th, and 21st pulse, the created skyrmions are 1, 5, and 19, respectively. The depth and width of the notch are 280 nm and 190 nm, respectively. The orange circle highlights a skyrmion at the corner. **b** The number of created skyrmions as a function of pulse numbers. Inset: Schematic plots of the experimental setup (upper left) and a magnetic skyrmion (lower right). **c** The average numbers of created skyrmions per current pulse as a function of current density. The creation rate is asymmetric with respect to the current direction, especially in the STT-dominated region. The scale bar in (**a**) is 400 nm. **d** Unidirectional skyrmion creation in the STT-dominated regime. Skyrmions with the topological charge of $Q = -1$ ($Q = +1$) are created on the right (left) side of the notch under a negative (positive) $j$ and a positive (negative) external field $B$. No skyrmion is created under the combinations $j > 0$, $B > 0$, and $j < 0$, $B < 0$. The unidirectional skyrmion creation originates from the unique direction of the spin precession that breaks the reflection symmetry. $\odot$ and .. stand for the upward and outward directions of the external magnetic field, respectively. The amplitude of the magnetic field is $B = 70$ mT, and the current density is $|j| = 4.26 \times 10^{10}$ A · m$^{-2}$ and pulse width 20 ns.

direction under a negative current with pulse width of 80 ns and $j \sim -3.48 \times 10^{10}$ A · m$^{-2}$ (Supplementary Movie 5 for $j \sim -2 \times 10^{10}$ A · m$^{-2}$). The transverse motion along the $+y$ direction, i.e., skyrmion Hall effect (quantified as $\tan \theta_h = v_y/v_x$) is observed, as depicted in Fig. 3a. The trajectory (Fig. 3e) shows approximately linear behavior as predicted in theory. The pinning effect and thermal fluctuation can reasonably explain the deviation from the linearity. The longitudinal velocity $v_x$ is always antiparallel to the current flow while the transverse velocity $v_y$ is related to the topological charge $Q$ (Supplementary Note I and Fig. 3). Therefore, the magnetic skyrmion moves in the opposite direction when a positive current is applied, and a reversal sign of $Q$ only changes the direction of velocity $v_y$ (Fig. 3b, d).

The current-driven skyrmions motion can also be observed in skyrmion cluster states (Fig. 4). Figure 4a, b show the collective motion of skyrmion clusters with $N_s = 4$ and $N_s = 26$, respectively. Both the velocity and skyrmion Hall angle are similar to those of the single skyrmion, which is because the shape factor $\eta$ of a cluster scales linearly with $N_s$ (see Supplementary Note I). However, the trajectories' deviation from straight lines is significantly suppressed with the increased number of skyrmions in the cluster states (Supplementary Fig. S7). The collective motions of two skyrmion clusters are possible, as shown in Supplementary Fig. S8 with $N_s = 11$ and $N_s = 21$, where the distance between the two clusters remains constant during the

motion. Interestingly, the skyrmion clusters can even steadily pass through a defect without noticeable deformation (Supplementary Fig. S8 and Movie 6).

Based on the trajectories of magnetic skyrmions under varied current densities, the current-density-dependent skyrmion velocities are summarized in Fig. 3e. To minimize the uncertainty, skyrmion clusters with the number of $N_s \sim 20$ is selected therein. The predicted linear relationship[32] between the skyrmion velocity and current density is obtained. Moreover, the estimated spin polarization of current for $Co_8Zn_{10}Mn_2$ is $P \sim 0.57$ (see Methods), which is two times larger than that for FeGe[12] ($P \sim 0.27$), resulting in a comparable efficiency (defined as $\varepsilon = v_x/j$) to the reported record using SOT mechanism[13,16]. Below a low critical current density $j_{c1} = 1.0 \times 10^{10}$ A · m$^{-2}$, magnetic skyrmions are static. This critical current density is directly related to the pinning forces arising from the disorder or impurity[32]. In addition, the critical current density depends on the pulse width as well. It decays exponentially with respect to the pulse width and reduces to $\sim 5 \times 10^9$ A · m$^{-2}$ at the pulse width of 200 ns (Supplementary Fig. S9). Above the critical density $j_{c2} = 3.5 \times 10^{10}$ A · m$^{-2}$, skyrmions are dynamically created and annihilated due to the combined effect of STT and the Joule heating by current pulses[27].

Figure 3f depicts an inverse relationship between the skyrmion Hall angle ($\theta_h$) and the current density. In a defect-free system, the skyrmion Hall angle $\theta_h$ should be constant. However, in

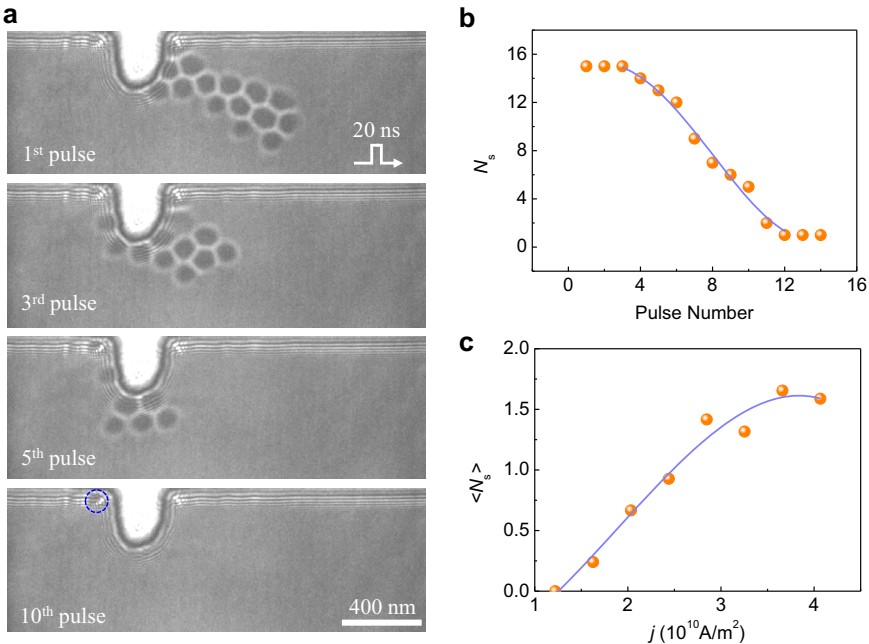

**Fig. 2 Skyrmion annihilation with a notch. a** A skyrmion cluster is pushed towards the notch using current pulses and absorbed by the notch gradually. The snapshots were taken under the defocus of 1 mm. The external field B = 70 mT, the pulse width is 20 ns and current density is $4.06 \times 10^{10}$ A · m$^{-2}$. **b** The number of skyrmions $N_s$ decreases as the current pulses are applied to the system. **c** The average number of erased skyrmion per pulse as a function of current density.

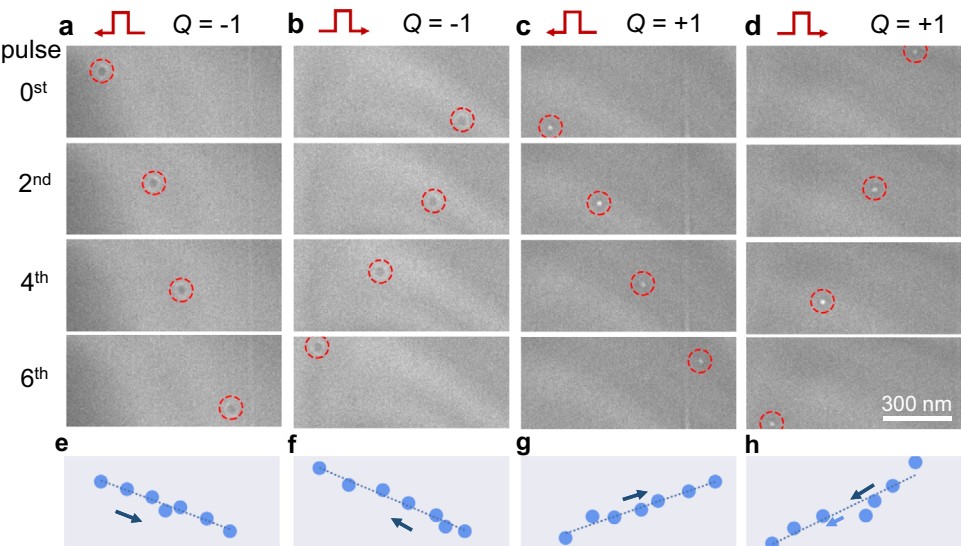

**Fig. 3 Lorentz TEM imaging of a single skyrmion motion driven by current pulses. a** A single skyrmion with $Q = -1$ moves forward with a negative $j < 0$. **b** The reversal motion of skyrmion with $Q = -1$ at $j > 0$. **c** A single skyrmion with $Q = +1$ moves forward at $j < 0$. **d** The reversal motion of skyrmion with $Q = +1$ at $j > 0$. The corresponding trajectories of the skyrmion motion are shown in (**e**–**h**). The fitted lines are shown to guide the eyes. The nonzero transverse motion in the $+y$ direction is characterized by the skyrmion Hall angle, which depends on the skyrmion number while not the current direction. A negative (positive) $Q$ is determined by the positive (negative) magnetic field at B = 94 mT. The current density is $|j| = 3.48 \times 10^{10}$ A · m$^{-2}$ and the pulse width is 80 ns. The scale bars are 300 nm.

natural materials, the defects-induced pinning force will give rise to a transverse motion of magnetic skyrmions, yielding an extrinsic skyrmion Hall effect[32] (Supplementary Note I and Supplementary Fig. S10). At low current density, a low drift velocity increases the scattering rate and thus results in a large skyrmion Hall angle (Supplementary Fig. S11 and Supplementary Movie 7). At higher current density, the scattering rate decreases, and the observed skyrmion Hall angle is close to the intrinsic value $\theta_h$, which is as low as ~15°. Our results are quite different

from previous studies in magnetic multilayers[16,33,34], where the skyrmion Hall angle shows a complicated relationship with current density. The underlying reason is that the skyrmion Hall angle therein is particularly susceptible to the change of radius under magnetic field and the deformation of the spin texture in the motion.

**Integrated operations of skyrmions.** At last, we demonstrate a combination of all creation, motion, and deletion operations on a

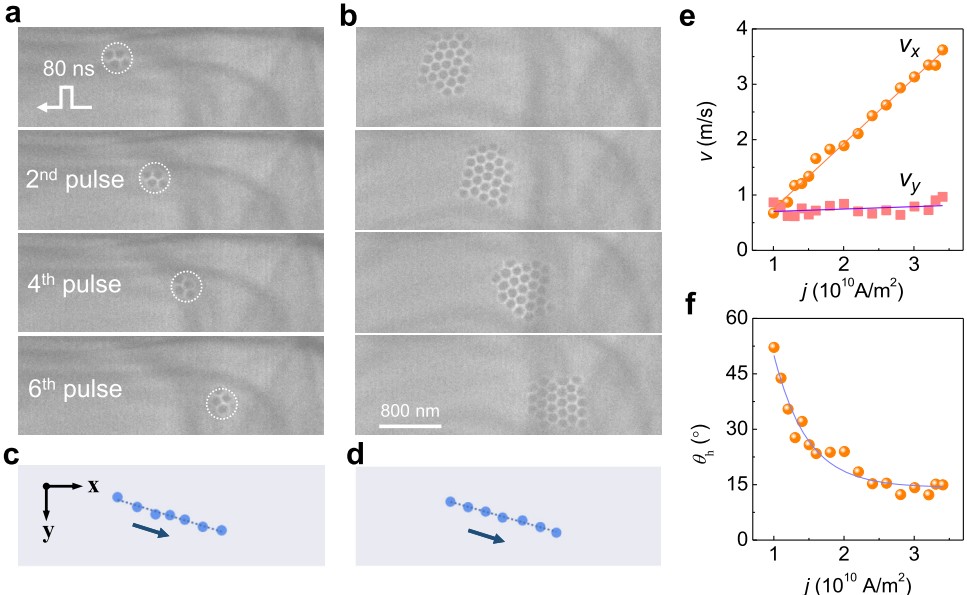

**Fig. 4 Spin transfer torque induced skyrmion clusters motion.** The sequences of images in (**a**) and (**b**) show the positions of skyrmions after applying the negative current pulses where the duration of each pulse is 80 ns, and the current density is $-2 \times 10^{10}$ A · m$^{-2}$. **a** A skyrmion cluster with $N_s = 4$. **b** A skyrmion cluster with $N_s = 26$. The corresponding trajectories of the skyrmion center are shown in (**c** and **d**) and fit well to straight lines. The negative pulses lead to positive displacements in the $+x$ direction independent of the skyrmion number. The skyrmion Hall angle characterizes the nonzero transverse motion in the $+y$ direction $\theta_h$ which depends on the skyrmion number and is unrelated to the current direction. The amplitude of the external field $B$ is 117 mT. **e** The skyrmion velocity as a function of the current density. The $x$-component velocity scales linearly with the current density. **f** The skyrmion Hall angle $\theta_h$ as a function of current density.

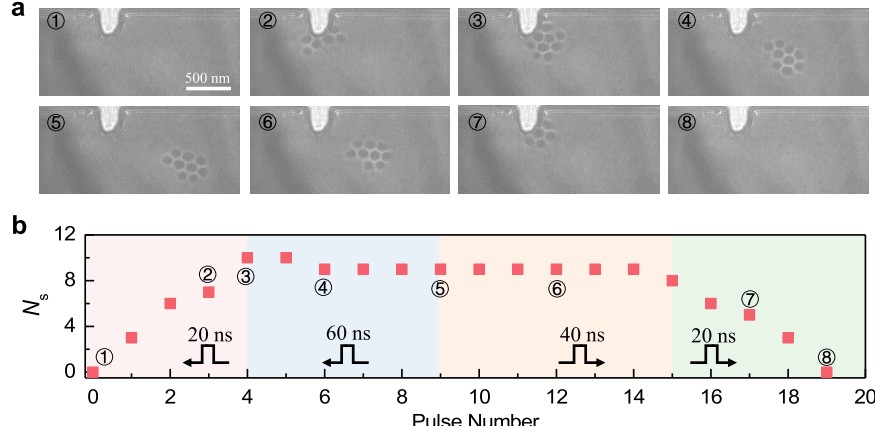

**Fig. 5 Electrically manipulation of a skyrmion cluster: a combination of creation, motion, and deletion. a** The snapshots of the skyrmion cluster on different stages. The skyrmion cluster is created in the first stage (①②③) and moves forward in the second stage (③④⑤). Then, the skyrmion is pushed back in the third stage (⑥) and is finally deleted in the fourth stage (⑦⑧). **b** The details of the current pulses and the number of skyrmions as a function of pulse number are plotted. The current densities used on these four stages are $-4.87 \times 10^{10}$ A · m$^{-2}$, $-2.03 \times 10^{10}$ A · m$^{-2}$, $2.03 \times 10^{10}$ A · m$^{-2}$ and $3.65 \times 10^{10}$ A · m$^{-2}$, respectively.

single device shown in Fig. 4 (see the details in Supplementary Movie 8 and Supplementary Fig. S12 for other datasets). In the first stage, a skyrmion cluster with ten skyrmions is created with the pulse width of 20 ns and $j \sim -4.86 \times 10^{10}$ A · m$^{-2}$ as shown in Fig. 4a. In the second stage, the skyrmion cluster is displaced by 680 nm after five pulses with the pulse width of 60 ns and $j \sim -2.03 \times 10^{10}$ A · m$^{-2}$. After that, the skyrmion cluster is pulled back with the pulse width of 60 ns at $j \sim 2.03 \times 10^{10}$ A · m$^{-2}$. In the final stage, the skyrmion cluster is deleted eventually with the pulse width of 20 ns at $j = 3.65 \times 10^{10}$ A · m$^{-2}$.

In this work, we show a proof-of-concept demonstration of necessary operations for skyrmion-based memory. In our experiments, this achievement of skyrmion creation, motion,

and deletion at room temperature enable the chiral magnets as a unique platform for skyrmion-based spintronic devices. Additionally, chiral magnets allow the coexistence of other exotic particle-like magnetic objects such as bobbers[35] and hopfions[36,37], making the versatile spintronic devices[38] based on three-dimensional spin textures possible.

## Methods

**Preparation of Co$_8$Zn$_{10}$Mn$_2$ crystals**. Polycrystalline samples of Co$_8$Zn$_{10}$Mn$_2$ crystals were synthesized by a high-temperature reaction method. Stoichiometric amounts of cobalt (Alfa Aesar, purity > 99.9%), zinc (Alfa Aesar, purity > 99.99%), and manganese (Alfa Aesar, purity > 99.95%) were loaded into a pure quartz tube and sealed under vacuum, heated to 1273 K for 24 h, followed by a slow cooling down to 1198 K, and then kept at this temperature for more than three days. After

that, the tube was quenched into cold water. Finally, a ball-shaped $Co_8Zn_{10}Mn_2$ alloy with a metallic luster was obtained.

**Fabrication of $Co_8Zn_{10}Mn_2$ micro-devices**. The $Co_8Zn_{10}Mn_2$ micro-devices suitable for TEM observation are fabricated from a polycrystal $Co_8Zn_{10}Mn_2$ alloy using the focus ion beam (FIB) dual-beam system (Helios NanoLab 600i; FEI) equipped with GIS and Omniprobe 200+ micromanipulator. A customized electrical chip with four Au electrodes was self-designed for the in-situ Lorentz transmission electron microscopy (TEM) experiment. The electrical TEM micro-devices are fabricated based on the conventional FIB lift-out method. The detailed procedures can be found in Supplementary Fig. S1 and Movie 1.

**Estimation of effective spin polarization for $Co_8Zn_{10}Mn_2$ and FeGe**. The effective spin polarization of current is established using the relation between the measured skyrmion velocity and the current density. Using the fitted slope $b = 1.18 \times 10^{-10}$ $m^3 \cdot A^{-1} \cdot s^{-1}$) and the estimated $M_s = 2.78 \times 10^5 A \cdot m^{-1}$ for $Co_8Zn_{10}Mn_2$ at 300K, the spin polarization of $Co_8Zn_{10}Mn_2$ is estimated with $P \approx 2eM_s b/g\mu_B \approx 0.57$. Similarly, the effective polarization for FeGe can be established as $P \sim 0.27$ using $M_s = 3.84 \times 10^5 A \cdot m^{-1}$ and $b = 4 \times 10^{-11} m^3 \cdot A^{-1} \cdot s^{-1}$, which is extracted from Fig. 3a in Ref. [12].

## Data availability

The data that support the plots provided in this paper and other findings of this study are available from the corresponding author upon reasonable request.

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

## Acknowledgements

H.D. acknowledges the financial support from the National Key R&D Program of China, Grant No. 2017YFA0303201; the Strategic Priority Research Program of Chinese Academy of Sciences, Grant No. XDB33030100; and the Equipment Development Project of Chinese Academy of Sciences, Grant No. YJKYYQ20180012; the Youth Innovation Promotion Association CAS No. 2015267; J.Z. was supported by the Office of Basic Energy Sciences, Division of Materials Sciences and Engineering, U.S. Department of Energy, under Award No. DE-SC0020221, and also supported by Alexander von Humboldt Foundation; D.S. acknowledges the financial support from the Chinese National Natural Science Foundation (52173215), the National Natural Science Fund for Excellent Young Scientists Fund Program (Overseas) and the Natural Science Foundation of Anhui Province for Excellent Young Scientist (2108085Y03).

## Author contributions

H.D. supervised the project. J.Z. conceived the theory. W.W., D.S., P.N., and H.D. conceived the experiments. W.-S.W. synthesized $Co_8Zn_{10}Mn_2$ crystals. W.W., D.S., H.D., and J.Z. prepared the manuscript. All authors discussed the results and contributed to the manuscript.

## Competing interests

The authors declare no competing interests.
