## [Peer Review File · Nature Communications]

Reviewers' Comments:

Reviewer #1:

Remarks to the Author:

Review: The manuscript by Wang et al., reports a study of Skyrmion creation, motion and annihilation measured by Lorentz TEM. Skyrmions are of interest for fundamental magnetism, as well as for realizing next-generation magnetic storage and computing technologies. Toward these ends, the electrically-driven nucleation, motion and annihilation of Skyrmions is desirable. This is an active and steadily progressing field, and there have been examples in recent years of each aspect of Skyrmion dynamics in the interfacial multilayer systems as well as in bulk chiral magnet materials. This manuscript is unusual in reporting electrically-driven nucleation, motion and annihilation of individual skyrmions as well as skyrmion clusters, all in the same system. The data are of very high quality and I support publication in Nature Communications, as these results will be of interest to a broad community of researchers.

One point which I thought could have used more discussion is the issue of the critical current density, which will be important for future applications. In particular, Yu and Tokura (Ref 21 in the manuscript, though the citation is incomplete) reported control over skyrmion dynamics with 10^8 A/m² in a nearly identical alloy (Co₈Zn₉Mn₃), which is two orders of magnitude lower than in the current study (10^{10} A/m²). If the authors could comment on their understanding of this difference, it would provide useful context for the readers.

As a point of clarity, I found the discussion on the unidirectionality confusing until I looked closely at the supplemental information (Fig. S2), which shows that the magnetic field direction is what breaks the symmetry with regards to current direction. This is an important point that could have been made clearer in the main text.

Reviewer #2:

None

Reviewer #3:

Remarks to the Author:

Wang et al., reports interesting in situ electron microscopy experiments on magnetic skyrmions in chiral magnets to demonstrate skyrmion creation, annihilation, and controlled motion by electrical current pulses. Using direct Lorentz electron microscopy, this study demonstrates such electrical manipulation of skyrmions in thin TEM foil magnets in a device structure. Results could be useful to design spintronic devices based on skyrmions and their clusters for skyrmion-based data storage and logic devices. In my view, this manuscript may be considered for publication. There are typos. English should be checked.

1. Authors claimed that they developed a new fabrication method of TEM samples in device structures. Video 1 in Supplementary shows a typical FIB-processing of TEM sample preparation. Various silicon nitride TEM chips like the one used in this study are available commercially (For example, TEMwindows.com). Authors should point out clearly what has been developed over conventional FIB lift-out process and use of silicon nitride chips with electrodes reported previously.

2. Authors pointed out that the notch behaves differently from "normal" sample edges in their experiments. They should describe more in detail the different behaviors in terms of skyrmion nucleation, attraction, pinning, repulsion, etc. For example, why skyrmion nucleation energy is lower at the notch compared to the "normal" sample edges under electric current flow? In addition, if these edges including the notch are made by FIB, how ion beam damage affects skyrmion nucleation and pinning?

3. Number of skyrmions created by number of electric current pulses do not consistent with the LTEM images in Fig. 1a and the plot shown in Fig. 1b. For example, for 21 current pulses, I only see 18 skyrmions in Fig. 1a. Similarly, in Fig. 1b, the number of skyrmions created by 20 pulses appears to be ~ 17 . Fig. 1a and Fig. 1b were from the same measurement? This brings a question on reproducibility of creation, annihilation, and controlled motion, which are main points in the manuscript. Most data only show single data points (plots in all Figures in the manuscript). I

understand this type of experiments could be challenging but multiple experiments are needed to confirm the linear creation dependence and so on?

4. Under the magnetic field (~ 70 mT) used in experiment, skyrmions are stable after creation? If they are meta-stable state, what are the typical lifetime of those created by electric current pulse?

5. In Fig. 3, it seems the topological Hall angles is about the same for different $Q=-1, -11,$ and -21 . The topological Hall angle should be linear-dependent on the topological charges?

6. Skyrmion creation in thermal regime: authors mention that with large current density, skyrmions are generated as the sample temperature increases above T_c and rapidly cooling down under external magnetic field ~ 70 mT. If sample was heated above T_c and went through field cooling, I think skyrmions are likely to form not only at the notch but also other areas in the sample, as shown in ref 27. However, Fig. S3 shows that skyrmions predominantly formed at the notches.

7. If there is no length limit, I would like to recommend Authors to add Supplementary data to the main text.

RE: manuscript NCOMMS-21-32854-T

"Electrical manipulation of skyrmions in a chiral magnet" by Wang *et al.*

Response to reviewers

We would like to express our gratitude to the reviewers for their careful reading, constructive comments and positive evaluations of our work. The revised version of the manuscript has been prepared based on all of the comments and suggestions from the reviewers. Please find the revised manuscript and Supplementary Information with changes highlighted and the point-by-point responses below for details. We look forward to continuing support from the reviewers towards publishing our results in *Nature Communications*.

The detailed point-to-point responses to all raised comments are shown below. All page numbers, references and figures refer to those in the revised manuscript and Supplementary Information. The changes are highlighted in blue in the revised manuscript and Supplementary Information.

Response to Reviewer #1:

Comment 1: The manuscript by Wang et al., reports a study of Skyrmion creation, motion and annihilation measured by Lorentz TEM. Skyrmions are of interest for fundamental magnetism, as well as for realizing next-generation magnetic storage and computing technologies. Toward these ends, the electrically-driven nucleation, motion and annihilation of Skyrmions is desirable. This is an active and steadily progressing field, and there have been examples in recent years of each aspect of Skyrmion dynamics in the interfacial multilayer systems as well as in bulk chiral magnet materials. This manuscript is unusual in reporting electrically-driven nucleation, motion and annihilation of individual skyrmions as well as skyrmion clusters, all in the same system. The data are of very high quality and I support publication in Nature Communications, as these results will be of interest to a broad community of researchers.

Response: We appreciate the reviewer for his/her high evaluation of our manuscript and strong support for its publication in Nature Communications.

Comment 2: One point which I thought could have used more discussion is the issue of the critical current density, which will be important for future applications. In particular, Yu and Tokura (Ref 21 in the manuscript, though the citation is incomplete) reported control over skyrmion dynamics with 10^8 A/m² in a nearly identical alloy (Co₈Zn₉Mn₃), which is two orders of magnitude lower than in the current study (10^{10} A/m²). If the authors could comment on their understanding of this difference, it would provide useful context for the readers.

Response: We thank the reviewer for pointing out the issue of the critical current density. The critical density depends on the pulse width. More precisely, a larger pulse width results in a lower critical current density. By varying the pulse width in our experiments, the pulse-width-dependent critical current density is obtained as shown in Fig. R1. The critical current density decays exponentially with respect to the pulse width. At the pulse width of 200 ns, the critical current density reduces to $\sim 5 \times 10^9$ A/m². Please note that a DC current is used in Ref. [21], and thus a lower critical density is expected.

Fig. R1. The experimental measured critical current density as a function of the pulse width. The yellow line is used to fit the data in the exponential form.

Revision: We have added Fig. R1 into the revised Supplementary Fig. S9 and some discussions of pulse-width-dependent critical current density in Lines 227-229 in the revised manuscript.

Lines 227-229: “In addition, the critical current density depends on the pulse width as well. It decays exponentially with respect to the pulse width and reduces to $\sim 5 \times 10^9$ A/m² at the pulse width of 200 ns (Supplementary Fig. S9).”

Comment 3: As a point of clarity, I found the discussion on the unidirectionality confusing until I looked closely at the supplemental information (Fig. S2), which shows that the magnetic field direction is what breaks the symmetry with regards to current direction. This is an important point that could have been made clearer in the main text.

Response: We appreciate the reviewer for the suggestion. The unidirectionality of the skyrmion creation means that only one direction can be used to create skyrmions once the notch and external fields are fixed. The underlying mechanism is that the direction of the spin precession in the LLG equation is unique, which breaks the reflection symmetry. This is also why the skyrmion can be erased when the current direction is reversed. To make this part clearer, we have merged the Supplementary Fig. S2 (*Unidirectional skyrmion creation in the STT-dominated regime*) into Fig. 1, as shown in Fig. R2, so the unidirectionality of the skyrmion creation can be clearly seen for the four combinations of current direction and applied field direction.

Fig. R2 | Skyrmion creation with a notch. **a**, A sequence of Lorentz TEM images of the skyrmion creation process after applying designated numbers of current pulses. After the 2nd, 6th, and 21st pulse, the created skyrmions are 1, 5, and 19, respectively. The depth and width of the notch are 280 nm and 190 nm, respectively. The orange circle highlights a skyrmion at the corner. **b**, The number of created skyrmions as a

function of pulse numbers. Inset: Schematic plots of the experimental setup (upper left) and a magnetic skyrmion (lower right). **c**, The average numbers of created skyrmions per current pulse as a function of current density. The creation rate is asymmetric with respect to the current direction, especially in the STT-dominated region. The scale bar in **a** is 400 nm. **d**. Unidirectional skyrmion creation in the STT-dominated regime. Skyrmions with the topological charge of $Q = -1$ ($Q = +1$) are created on the right (left) side of the notch under a negative (positive) j and a positive (negative) external field B . No skyrmion is created under the combinations $j > 0$, $B > 0$, and $j < 0$, $B < 0$. The unidirectional skyrmion creation originates from the unique direction of the spin precession that breaks the reflection symmetry. \odot and \otimes stand for the upward and outward directions of the external magnetic field, respectively. The amplitude of the magnetic field is $B = 70$ mT, and the current density is $|j| = 4.26 \times 10^{10}$ A/m² and pulse width 20 ns.

Revision: Figure 1 and its caption are updated with Fig. R2 in the revised manuscript. The discussion of unidirectionality is located accordingly in the Lines 122-128 in the revised manuscript.

Lines 122-128: “On the contrary, a positive current with the same current density failed to create skyrmions (Fig. 1d), indicating the asymmetric STT effects with regard to the current direction²⁴. This asymmetry of STT-induced skyrmion creation originates from the breakdown of reflection symmetry due to the unique direction of the spin precession²⁴. Consequently, skyrmions with $Q = +1$ could only be created with a positive current and a negative field (Fig. 1d).”

Response to Reviewer #3:

Wang et al., reports interesting in situ electron microscopy experiments on magnetic skyrmions in chiral magnets to demonstrate skyrmion creation, annihilation, and controlled motion by electrical current pulses. Using direct Lorentz electron microscopy, this study demonstrates such electrical manipulation of skyrmions in thin TEM foil magnets in a device structure. Results could be useful to design spintronic devices based on skyrmions and their clusters for skyrmion-based data storage and logic devices. In my view, this manuscript may be considered for publication.

Response: We appreciate the reviewer for the encouraging evaluation of our manuscript.

Comment 1: There are typos. English should be checked.

Response: We have checked our manuscript carefully and fixed all the typos accordingly in the revised version. English throughout the manuscript has been carefully improved as well.

Comment 2: Authors claimed that they developed a new fabrication method of TEM samples in device structures. Video 1 in Supplementary shows a typical FIB-processing of TEM sample preparation. Various silicon nitride TEM chips like the one used in this study are available commercially (For example,

TEMwindows.com). Authors should point out clearly what has been developed over conventional FIB lift-out process and use of silicon nitride chips with electrodes reported previously.

Response: We thank the reviewer's comment for providing the new information on commercial TEM chips. The silicon nitride chip used in our study is self-designed and customized for in-situ electrical experiments. Based on such a chip, the fabrication of the TEM microdevice is compatible with the conventional FIB lift-out method, making the whole process efficient. This is what we would like to highlight as "new efficient method", but we agree that it is actually similar to the commercial chips as the reviewer mentioned.

Revision: We have rewritten the statements in Lines 61-67 in the revised manuscript.

Lines 61-67: "A customized electrical chip with four Au electrodes was self-designed for the *in-situ* Lorentz transmission electron microscopy (TEM) experiment. The electrical TEM micro-devices are fabricated based on the conventional focus ion beam (FIB) lift-out method. The whole process is summarized in Supplementary Video 1 and Fig. S1. Based on this, the current-induced motion of a new type of multi-Q skyrmionic texture, termed as skyrmion bundle²², has been demonstrated in a chiral magnet FeGe at the low temperature of $T \sim 95$ K."

Comment 3: Authors pointed out that the notch behaves differently from "normal" sample edges in their experiments. They should describe more in detail the different behaviors in terms of skyrmion nucleation, attraction, pinning, repulsion, etc. For example, why skyrmion nucleation energy is lower at the notch compared to the "normal" sample edges under electric current flow? In addition, if these edges including the notch are made by FIB, how ion beam damage affects skyrmion nucleation and pinning?

Response: We thank the reviewer for his/her comments on the notches. The essential difference between the notch and normal edges is the spin texture. The notch has a large in-plane magnetization component that is perpendicular to the direction of electrical current, while it is parallel at the normal edges. It is this key difference that leads to easy skyrmion nucleation around the notch. As shown in Fig. R3 below (*Simulation of the skyrmion creation under a current pulse*), under the electric current, the spin textures could swell out and then nucleate a skyrmion due to the STT. Such process cannot happen at the normal edges.

The notch is fabricated by the FIB. In the experiments, we have used the low energy ion beam to reduce the damages as much as possible. Therefore, only a very thin amorphous layers (< 5 nm) is present at the edges. It does not influence the nucleation as the overall magnetism is still well kept at the notch.

Fig. R3 | Simulation of the skyrmion creation under a current pulse. a-f, Snapshots of the dynamical magnetization at selected times. A skyrmion with the topological charge of $Q = -1$ is created on the right side of the notch under a negative j and a positive external field B . In the simulation, $u = 50$ m/s is used.

Revision: We have added the Fig. R3 into the Supplementary Fig. S3. The discussions of skyrmion dynamics at the notch and normal edges have been added in Lines 104-105 and Lines 112-116 in the revised manuscript.

Lines 104-105: “The created skyrmion is attached to the notch, indicating the attractive interaction between the skyrmion and the notch.”

Lines 112-116: “The spin textures of the notch play an essential role in creating the skyrmions. The notch has a sizeable in-plane magnetization component that is perpendicular to the direction of electrical current, while it is parallel at the normal edges. Under the electrical current, the spin textures could swell out and then nucleate a skyrmion due to the STT, but absent at the regular edges (Supplementary Fig. S3). Therefore, the skyrmion nucleation energy is lower at the notch than the “normal” sample edges.”

Comment 4: Number of skyrmions created by number of electric current pulses do not consistent with the LTEM images in Fig. 1a and the plot shown in Fig. 1b. For example, for 21 current pulses, I only see 18 skyrmions in Fig. 1a. Similarly, in Fig. 1b, the number of skyrmions created by 20 pulses appears to be ~ 17 . Fig. 1a and Fig. 1b were from the same measurement? This brings a question on reproducibility of creation, annihilation, and controlled motion, which are main points in the manuscript. Most data only show single data points (plots in all Figures in the manuscript). I understand this type of experiments could be challenging but multiple experiments are needed to confirm the linear creation dependence and so on?

Response: We appreciate the reviewer for carefully checking our manuscript. We are sorry that the statement is inconsistent here. The data shown in Fig. 1b were extracted

from Fig. 1a and they describe the same measurement. Actually, there are 19 skyrmions for the 21st current pulses (marked with orange points in the following Fig. R4) and 18 skyrmions for the 20th current pulses. Particularly, we have used the orange circle to mark the skyrmion at the upper left corner, which is not obvious in the Fig. R4.

Fig. R4. 19 skyrmions are created after the 21th current pulses.

Regarding to the reproducibility of creation, deletion and motion, we have included additional experimental data (Fig. R5 for skyrmion creation and Fig. R6 for the combined skyrmion operation) to support our conclusion in the revised supplementary information.

Fig. R5 | Skyrmion creation with a notch. **a-c**, The sequence of Lorentz TEM images of skyrmion creation at $j = -4.06 \times 10^{10} \text{ A/m}^2$, $j = -4.46 \times 10^{10} \text{ A/m}^2$ and $j = -4.87 \times 10^{10} \text{ A/m}^2$, respectively. **d-f**, The number of created skyrmions as a function of applied pulses for these three current densities, as depicted in **a-c**. The linearity between the number of created skyrmion and pulses are observed for various current densities. The scale bar is 400 nm.

Fig. R6 | Electrically manipulation of a skyrmion cluster. **a**, The snapshots of the skyrmion cluster on different stages. The skyrmion cluster is created in the first stage (①②③) and moves forward in the second stage (③④⑤). The skyrmion then is pushed back on the third stage (⑥) and is finally deleted in the fourth stage (⑦⑧). **b**, The details of the current pulses and the number of skyrmions as a function of pulse number are plotted. The current densities used on these four stages are $-4.87 \times 10^{10} \text{ A/m}^2$, $-2.03 \times 10^{10} \text{ A/m}^2$, $2.03 \times 10^{10} \text{ A/m}^2$ and $2.03 \times 10^{10} \text{ A/m}^2$, respectively. The scale bar is 400 nm.

Revision: The orange circle is marked in Fig. 1a to highlight the skyrmion at the corner. We have included three additional sets of data (Fig. R5) in the revised Supplementary Fig.S3 to show the linearity of the creation process. We have included an additional set of data (Fig. R6) in the revised Supplementary Fig.S12 to show the combined operations of creation, motion, and deletion.

Comment 5: Under the magnetic field ($\sim 70 \text{ mT}$) used in experiment, skyrmions are stable after creation? If they are meta-stable state, what are the typical lifetime of those created by electric current pulse?

Response: Under the magnetic field ($\sim 70 \text{ mT}$), the zero temperature ground state is the conical phase. The created single skyrmion state is a meta-stable state. We have not measured the lifetime of the created skyrmions very accurately. However, according to the experiment, the created skyrmion can exist for at least 10 hours. Sometimes after finishing one day's experiments, we didn't erase the created

skyrmions. The following day when the experiment resumed, we could still see the skyrmions at the notch with their number unchanged.

Comment 6: In Fig. 3, it seems the topological Hall angles is about the same for different $Q=-1, -11, \text{ and } -21$. The topological Hall angle should be linear-dependent on the topological charges?

Response: We appreciate the reviewer for raising this issue. The Hall angle of a skyrmion cluster is actually independent of the topological charges Q . The skyrmion Hall angle can be calculated as (see also Supplementary Note I),

$$\tan\theta_h = \frac{v_y}{v_x} = \frac{(\alpha-\beta)\eta Q}{Q^2+\eta^2\alpha\beta} \approx \frac{\eta}{Q}(\alpha - \beta) \quad (\text{Eq. R1})$$

where $\eta \propto \int d^2r(\nabla m)^2$ represents the shape factor of the skyrmion cluster. For a skyrmion cluster with N_s skyrmions, we have $\eta = N_s \eta_1$ where η_1 is the shape factor of a single skyrmion. So we have

$$\tan\theta_h = \frac{v_y}{v_x} \approx \frac{\eta}{Q}(\alpha - \beta) = \frac{\eta_1}{Q_1}(\alpha - \beta) \quad (\text{Eq. R2})$$

where $Q_1 = \pm 1$ is the topological charge for a single skyrmion. Therefore, the skyrmion Hall angle is independent of the cluster size N_s .

Revision: The equation Eq.(S8) has been updated as Eq. R2 in the revised Supplementary Note I. We have also added more discussion to address the Q -independent skyrmion Hall angle, Lines 207-209, in the revised manuscript.

Lines 207-209: “Fig.4a and Fig.4b show the collective motion of skyrmion clusters with $N_s = 4$ and $N_s = 26$, respectively. Both the velocity and skyrmion Hall angle are similar to those of the single skyrmion, which is because the shape factor η of a cluster scales linearly with N_s (see Supplementary Note I).”

Comment 7: Skyrmion creation in thermal regime: authors mention that with large current density, skyrmions are generated as the sample temperature increases above T_c and rapidly cooling down under external magnetic field ~ 70 mT. If sample was heated above T_c and went through field cooling, I think skyrmions are likely to form not only at the notch but also other areas in the sample, as shown in ref 27. However, Fig. S3 shows that skyrmions predominantly formed at the notches.

Response: We appreciate the reviewer for carefully checking our manuscript. We agree with the reviewer that the skyrmions should form anywhere in a uniform nanoribbon if the current is sufficiently high as shown in Ref.[27]. However, in the presence of a notch, the current density is nonuniform. A typical distribution of the current in the presence of a notch is shown in Fig. R7a. The area closed to the notch is hotter than other areas due to the Joule heating. Therefore, skyrmions are predominantly formed at the notches for a moderate current density, such as $|j| =$

$4.87 \times 10^{10} \text{ A/m}^2$ as shown in Fig. S4. However, when the current density increases further, the area of forming skyrmions increase rapidly and skyrmions appear in other areas, as shown in Fig. R7b at a current density of $|j| = 5.68 \times 10^{10} \text{ A/m}^2$. It can be seen that the skyrmions are formed not only at the notch but also in other areas.

Fig. R7. **a.** The spatial distribution of the amplitude of current density in the presence of a notch. **b.** The generated skyrmion after applying pulses with the current density of $j = -5.68 \times 10^{10} \text{ A/m}^2$.

Comment 8: If there is no length limit, I would like to recommend Authors to add Supplementary data to the main text.

Response: We thank the reviewer for the suggestion. We have merged the Supplementary Fig. S2 (*The unidirectional skyrmion creation in the STT-dominated regime*) into Fig. 1 as mentioned above. We have also moved the Supplementary Fig. S6 (*The single skyrmion dynamics under various external fields and different currents*) into the main text as Fig. 3 as shown below.

Fig. 3 | Lorentz TEM imaging of a single skyrmion motion driven by current pulses. **a**, A single skyrmion with $Q = -1$ moves forward with a negative $j < 0$. **b**, The reversal motion of skyrmion with $Q = -1$ at $j > 0$. **c**, A single skyrmion with $Q = +1$ moves forward at $j < 0$. **d**, The reversal motion of skyrmion with $Q = +1$ at $j > 0$. The corresponding trajectories of the skyrmion motion are shown in **e** to **h**. The fitted lines are shown to guide the eyes. The nonzero transverse motion in the $+y$ direction is characterized by the skyrmion Hall angle, which depends on the skyrmion number while not the current direction. A negative (positive) Q is determined by the positive (negative) magnetic field at $B = 94$ mT. The current density is $|j| = 3.48 \times 10^{10}$ A/m² and the pulse width is 80 ns. The scale bars are 300 nm.

Reviewers' Comments:

Reviewer #1:

Remarks to the Author:

I have read the authors' responses to both reviews and also the revised manuscript. In my view the authors have adequately responded to the specific points raised in both reviews and the manuscript is ready for publication. Some of the new changes could benefit from more editing however. For example, the addition of the sample holder discussion on pg 1 of the main text (lines 61-67) seems out of place.

Reviewer #3:

Remarks to the Author:

My comments are addressed sufficiently. Therefore, I would like to recommend this revised manuscript for publication.

Response to Reviewer #1:

Comment 1: I have read the authors' responses to both reviews and also the revised manuscript. In my view the authors have adequately responded to the specific points raised in both reviews and the manuscript is ready for publication.

Response: We appreciate the reviewer for his/her support of our manuscript for publication in Nature Communications.

Comment 2: Some of the new changes could benefit from more editing however. For example, the addition of the sample holder discussion on page 1 of the main text (lines 61-67) seems out of place.

Response: The discussion of sample holder and device preparation has been moved to the **Methods** part in the revised manuscript.

Response to Reviewer #3:

Comment 1: My comments are addressed sufficiently. Therefore, I would like to recommend this revised manuscript for publication.

Response: We appreciate the reviewer for his/her support of our manuscript for publication in Nature Communications.